# Rethinking Medical Report Generation:
# Disease Revealing Enhancement with Knowledge Graph

**Yixin Wang** [* 1]  **Zihao Lin** [* 2]  **Haoyu Dong** [* 3]

## Abstract

Knowledge Graph (KG) plays a crucial role in Medical Report Generation (MRG) because it reveals the relations among diseases and thus can be utilized to guide the generation process. However, constructing a comprehensive KG is labor-intensive and its applications on the MRG process are under-explored. In this study, we establish a complete KG on chest X-ray imaging that includes 137 types of diseases and abnormalities. Based on this KG, we find that the current MRG data sets exhibit a long-tailed problem in disease distribution. To mitigate this problem, we introduce a novel augmentation strategy that enhances the representation of disease types in the tail-end of the distribution. We further design a two-stage MRG approach, where a classifier is first trained to detect whether the input images exhibit any abnormalities. The classified images are then independently fed into two transformer-based generators, namely, "disease-specific generator" and "disease-free generator" to generate the corresponding reports. To enhance the clinical evaluation of whether the generated reports correctly describe the diseases appearing in the input image, we propose diverse sensitivity (DS), a new metric that checks whether generated diseases match ground truth and measures the diversity of all generated diseases. Results show that the proposed two-stage generation framework and augmentation strategies improve DS by a considerable margin, indicating a notable reduction in the long-tailed problem associated with under-represented diseases.

---

[*]Equal contribution [1]Department of Bioengineering, Stanford University, Stanford, CA, USA [2]Department of Computer Science, Virginia Tech, Blacksburg, VA, USA [3]Department of Electrical and Computer Engineering, Duke University, Durham, NC, USA. Correspondence to: Yixin Wang <yxinwang@stanford.edu>.

*Proceedings of the 40th International Conference on Machine Learning*, Honolulu, Hawaii, USA. PMLR 202, 2023. Copyright 2023 by the author(s).

## 1. Introduction

Chest radiography is one of the most common and effective imaging examinations used in clinical practice for diagnosing diseases and evaluating health risks. The obtained images generally require medical reports with comprehensive interpretation written by qualified physicians or pathologists, which can be time-consuming and requires expertise. With the advancement in deep learning (DL) algorithms, automatic medical report generation (MRG) has been widely explored and achieved significant performance (Jing et al., 2018; Wang et al., 2018; Xue et al., 2018; Li et al., 2018; Boag et al., 2020; Chen et al., 2020; Liu et al., 2021; Wang et al., 2021; Chen et al., 2021; Liu et al., 2019; Wang et al., 2022; Yang et al., 2023). These DL-based systems analyze the chest images and automatically generate a descriptive report outlining the findings. However, these methods are primarily designed to optimize the performance of matching generated N-gram to ground truth reports, rather than focusing on aligning generated medical attributes, *i.e.*, abnormalities or diseases with the actual reports, which is more important when assessing the clinical utility of a generation algorithm. While some researchers (Irvin et al., 2019; Harzig et al., 2019; Zhang et al., 2020) propose disease labeling tools or build disease knowledge graphs to aid in evaluating the reports, their KG contains limited disease types and they only consider report-level n-gram matching accuracy, which is a coarse reflection of the medical attributes.

To address these problems, we construct a large KG with 137 types of chest diseases based on two widely used chest X-ray datasets, IU-Xray (Demner-Fushman et al., 2016) and MIMIC-CXR (Johnson et al., 2019) (See Section 2 for details). Utilizing the diseases from this KG, a rule-based criterion is adopted to make a detailed statistical analysis on the appearing diseases and abnormalities in IU-Xray. As depicted in Figure 1(a), across all reports in the data set, the frequency of sentences indicating normal results (no diseases or abnormalities) is three times greater than those indicating the presence of at least one disease or abnormality. Moreover, the number of sentences with common diseases (occurrences greater than 20) is almost 4 times of those with uncommon diseases (occurrences less than 20). The

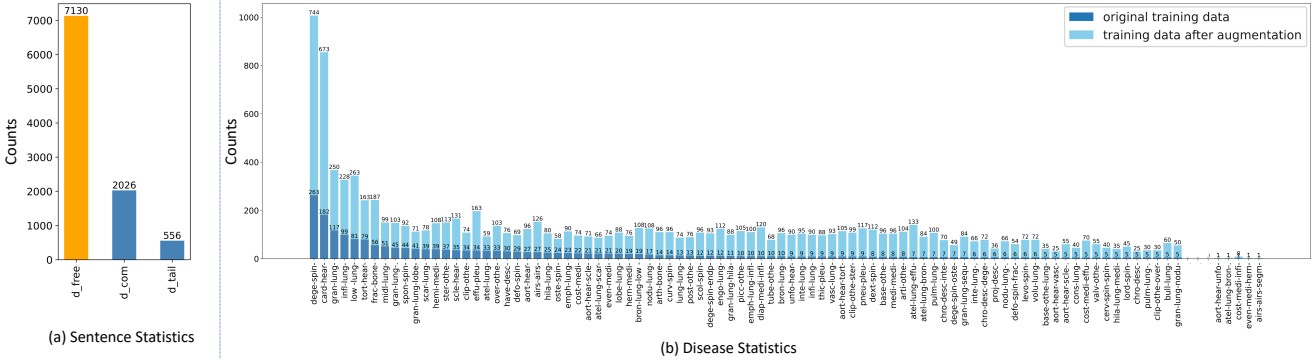

*Figure 1.* Illustration of counts of labeled sentences and disease keywords in IU-Xray. Part (a) shows the count of sentences that have common diseases (d_com), uncommon diseases (d_tail), or do not have diseases (d_free). Part (b) shows parts of distributions of diseases and abnormalities in original (dark blue bars) / augmented (light blue bars) training data. The lower / upper numbers are the occurrence counts of specific disease keywords in original / augmented training data.

frequency of occurrence for each disease keyword is further highlighted in Figure 1(b), which exhibits a long-tailed distribution of the disease classes in the data set. In the original training data (dark blue bars), only three diseases appear more than 100 times and 65.7% diseases appear less than 10 times in IU-Xray, which shows that several common diseases dominate but rarer ones are under-represented. In response, we design a two-stage generation approach to reduce the bias towards generating "disease-free" reports instead of "disease-specific" reports, i.e., reports that contain at least one disease or abnormality. We further alleviate the long-tailed distribution issue by expanding the distribution of the disease classes through a designed disease augmentation strategy. According to our statistics on the augmented training data (light blue bars in Figure 1(b)), the overall frequency of uncommon diseases in the original data set increases from 37.6% to 55.5%, while the common diseases see a decrease in overall frequency from 62.4% to 44.5%.

Moreover, when evaluating generated reports, more emphasis should be placed on clinic-efficacy information. Previous works employ the commonly used N-gram evaluation metrics from image captioning tasks, such as BLEU-N (Papineni et al., 2002). However, these metrics do not necessarily reflect the clinical quality of the diagnostic reports, such as the accuracy of the specific diseases. In our experiments, as well as in previous studies (Harzig et al., 2019), it has been observed that with an imbalanced data set, models tend to achieve the highest BLEU score when generating repetitive sentences that most frequently appear in the training set. Li et al. (Li et al., 2021) argue that the quality of medical reports largely depends on the accurate detection of positive disease keywords. Therefore, they employ several human evaluations as additional measurements. Nevertheless, implementing this evaluation requires significant expert efforts and is prone to subjectivity and variability. Based

on KG, we propose a new evaluation metric, diverse sensitivity (DS), to assess the model's ability to generate reports containing special diseases, which concentrates more on clinical-relevant texts. Our KG and codes will be available at https://github.com/Wangyixinxin/MRG-KG.

Our contributions are as follows:

- A complete knowledge graph with 8 disease categories and 137 diseases or abnormalities of chest radiographs is built based on accurate and detailed disease classification.
- A novel augmentation strategy is proposed to address the long-tailed problems in chest X-ray data sets.
- An effective two-stage MRG approach is designed to separately handle normal and abnormal images, generating texts more specific to the identified diseases.
- A KG-based evaluation metric, DS, is further proposed to assess the quality of generated reports, prioritizing the accuracy of disease-relevant attributes.

## 2. Knowledge Graph

Starting from (Zhang et al., 2020), several works have demonstrated the effectiveness of KG on chest report generation (Li et al., 2019; Liu et al., 2021; Zhang et al., 2020). The existing KG, which includes the most common diseases or abnormalities (Zhang et al., 2020), consists of 7 organs with 18 corresponding diseases, along with "normal" and "other findings". However, this KG lacks comprehensiveness as it omits many common diseases such as "calcification", "spine degenerative", and "lung consolidation". The restriction in disease types places a limitation on the model's capacity to learn about the relationships between diseases, resulting in a lack of clinical depth. For example, lung opacity can be divided into categories like "nodular opacity", "lobe opacity", and "hilar opacity". Besides, identical abnormali-

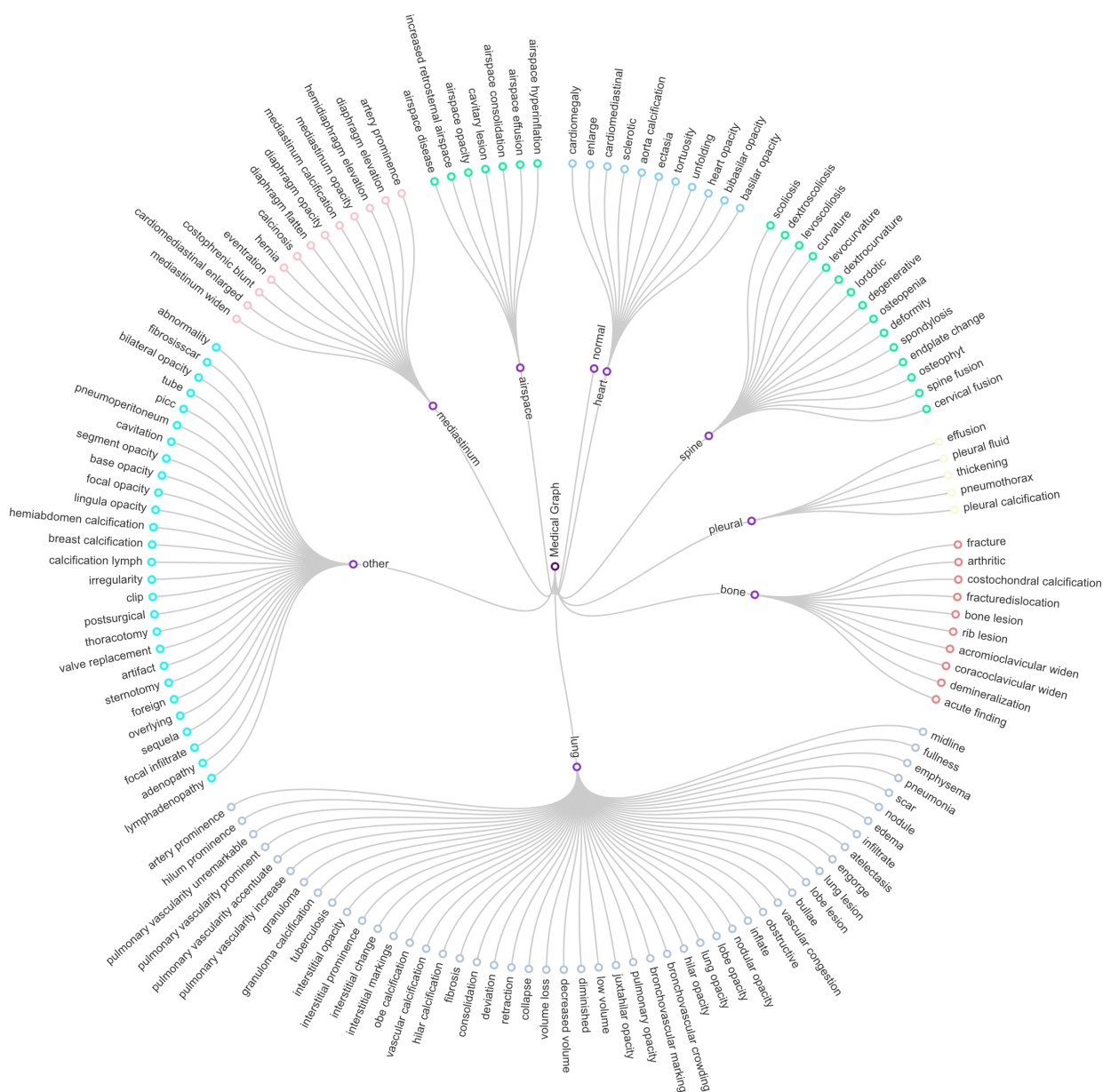

*Figure 2.* An illustration of our proposed knowledge graph, which contains "normal" and 8 disease categories including 7 organs and an "other" category. Each category further branches out into its corresponding specific diseases.

ties can appear in different organs, such as "lung opacity", "diaphragm opacity" and "airspace opacity". Lastly, the current KG does not account for several rare diseases or anomalies, leaving them unclassified. To overcome these limitations, we extend the knowledge graph by adding more diseases based on IU-Xray (Demner-Fushman et al., 2016) and MIMIC-CXR (Johnson et al., 2019). Figure 2 depicts a partial representation of our proposed knowledge graph. In our work, we retain the current seven organ categories while supplementing them with additional diseases. We also introduce another new category "other", which contains abnormalities, such as "tube" and "sternotomy" that do not belong to any of the seven organs. While constructing this KG, we also take into account the synonyms and variations of each specific disease, leading to a comprehensive representation of 137 disease types. These will be leveraged in our training approach (See Section 3) and evaluation metrics (See Section 4).

Based on the knowledge graph, we build a rule-based criterion to classify diagnostic reports. Firstly, each word will be replaced by its synonyms through a pre-defined synonyms pool. Then, each sentence in the report will be labeled by a concatenation of "diseases-organs" pairs if it includes keywords from the KG or "normal" class otherwise. For example, the sentence "there are low lung volumes with bronchovascular crowding" will be labeled as "bronchovascular crowding-lung-low volume-lung". A report is labeled as "disease-free" if all its sentences are labeled as "normal", otherwise it's marked as "disease-specific". These report labels will be utilized to train a classifier in our proposed two-stage generation approach.

## 3. Two-Stage Generation Approach

Figure 1 illustrates an imbalance distribution within the IU-Xray dataset between the number of sentences that indicates the presence or absence of diseases, along with a long-tail issue in the disease distributions. To address these issues, we propose two solutions: firstly, a novel two-stage pipeline including an image classifier and two generation networks with identical structures. These networks are trained with "disease-free" and "disease-specific" data separately (Section 3.1). Secondly, a disease-specific augmentation strategy to alleviate the imbalanced distribution of disease data (Section 3.2).

### 3.1. Training and Inference Stage

To address the dominance of normal findings in the data, we propose a two-stage approach. During the training phase, we leverage the available ground-truth reports to segregate the training data into the defined two classes, *i.e.*, "disease-free" and "disease-specific". Following this strategy, the images corresponding to each report, paired with their respective labels, are leveraged to train an image classifier, ResNet101 (He et al., 2016), with standard cross-entropy loss to detect if an input image contains diseases. In parallel, we employ two generative models for report generation: a "disease-free generator" and a "disease-specific generator", each trained on data from their respective classes. Both generators utilize the same architectural design based on R2Gen (Chen et al., 2020), one of the most popular approaches for MRG. Specifically, given a radiology image as an input, a visual extractor is trained to extract related features. Subsequently, a transformer encoder and a transformer decoder, both consisting of a multi-head self-attention and a multi-head cross-attention module, are further employed to generate long reports. During the inference stage, a two-stage approach is adopted, where an input image is first fed to the image classifier to distinguish whether it contains any disease or abnormality, and then the corresponding generator is chosen to generate the diagnostic report in the second stage.

Although the two-stage strategy can improve the ability of the generator to specifically generate "disease-specific" reports, there is still an inherent challenge of data imbalance which biases the model towards producing reports of the most dominant diseases found in the training data. With our disease KG, we further propose a novel data augmentation method to mitigate the disease imbalance issue.

### 3.2. Disease-Specific Augmentation

The first step of our augmentation strategy is to create a key-value pool of disease sentences, where the keys represent sentence labels (See Section 2) which are a concatenation of "diseases-organs" pairs such as "opacity-lung", and values include all unique-format sentences under this label such as "The lung is opacity" and "This patient has lung opacity". We define the label count as the number of unique-format sentences for each sentence label. A higher label count indicates more sentence variations that describe that label, which is easier to perform disease augmentation through random substitution. Therefore, we define a count interval [5, 100] by omitting sentence labels with a label count less than 5 or more than 500. Starting from the label with the fewest unique-format sentences in this interval, we first find all diagnostic reports that contain sentences under this sentence label. For each report, we substitute the sentence under this sentence label with another format from the key-value pool and repeat this operation for all reports. For example, if 5 distinct sentences belong to a particular label, the proposed augmentation strategy will generate $5 \times (5 - 1) = 20$ additional reports. Given that a report might contain multiple disease sentences, this augmentation process could inadvertently boost the frequency of various diseases concurrently. To moderate this undesired effect, we update the statistics of disease labels after each round of augmentation and find the next least frequent disease that has not been

augmented before. Figure 1 (b) indicates that the applied augmentation strategy successfully evens out the distribution of diseases, especially reducing the long-tail problem. It is noted that although the augmentation strategy increases the occurrences of all types of diseases, it prioritizes the occurrence of diseases in the tailed population.

## 4. Evaluation Metric

Common evaluation metrics, including BLEU (Papineni et al., 2002), ROUGE (Lin, 2004), METEOR (Banerjee & Lavie, 2005), etc., fail to consider whether the generated reports describe the diseases appearing in the input image. The accurate description of disease keywords is the main criterion for radiologists to decide whether to use the generated reports. Therefore, based on our proposed KG, we introduce a new evaluation metric, Diverse Sensitivity (DS), that evaluates whether the diseases identified in the ground-truth report are also accurately depicted in the generated report. Firstly, we consider a generated report to be correct *iff* it depicts at least one disease that appears in the ground truth. The Sensitivity (Sen.) is defined as $Sen. = \frac{TP}{TP+FN}$, where $TP$ and $FN$ stand for true positive and false negative respectively. However, due to the long-tail disease distribution, the network is able to achieve a high sensitivity if all the generated reports contain the most common disease. Thus, we propose a different metric, Diversity (Div.), to account for the variability during generation. Div. is defined as the ratio between the number of uniquely generated disease types and the number of total disease types. Lastly, DS is a harmonic mean of Sen. and Div., *i.e.*, $DS = 2 \times \frac{Sen. \times Div.}{Sen. + Div.}$. Since DS focuses on evaluating "disease-specific" sentences, we also introduce Diagnostic Odds Ratio (DOR), similar to the concept defined in (Glas et al., 2003). This complementary metric evaluates the model's ability to generate correct "disease-free" reports. Formally, $DOR = \frac{TP \times TN}{FP \times FN}$, where $TN$ and $FP$ stand for true negative and false positive respectively. DS and DOR are considered jointly to evaluate the clinical efficacy of a generation model.

## 5. Experiment

### 5.1. Datasets and Implementation

In the experiments, we adopt IU X-ray (Demner-Fushman et al., 2016) which consists of 7,470 chest X-ray images with 3,955 radiology reports. Each report is paired with two associated images - a frontal and a lateral view. This dataset is split into train/validation/test set by 7:1:2, following R2Gen (Chen et al., 2020). Model selection is based on the best DS score on the validation set and we report its performance on the test set. For a fair comparison, we keep all experimental settings consistent with those used in the R2Gen (Chen et al., 2020).

### 5.2. Comparison Results

#### 5.2.1. QUANTITATIVE RESULTS.

We evaluate the effectiveness of our proposed method as compared to R2Gen (Chen et al., 2020) using DS and DOR. We also include Sensitivity (Sen.) and Diversity (Div.) in our comparison for reference. As shown in Table 1, our method achieves a DS score of 0.1902 and a DOR score of 0.5138, outperforming R2Gen by a large margin. This improvement on these two clinical-relevant metrics implies a greater applicability of our method in real-world clinical settings.

| Method | DOR | DS | Sen. | Div. |
|---|---|---|---|---|
| R2Gen (Chen et al., 2020) | 0.2911 | 0.1523 | 0.0932 | 0.4153 |
| Two-Stage + Aug. (Ours) | **0.5138** | **0.1902** | 0.1220 | 0.4305 |
| Two-Stage | 0.4223 | 0.1634 | 0.1034 | 0.3898 |
| Disease-Specific Only | 0 | 0.1955 | 0.1305 | 0.3898 |
| R2Gen* (Chen et al., 2020) | 0.4366 | 0.0324 | 0.0186 | 0.1220 |

Table 1. Comparison results. R2Gen* means the best model under BLEU.

We further investigate the effect of augmentation and the two-stage generation process in Table 1. The observed decrease in DOR to 0.4223 and DS to 0.1634, when the low-frequency diseases are not augmented, emphasizes the significance of ensuring a balanced disease distribution in the data set. Given our objective to improve the correct generation of disease sentences, we compare the two-stage generation model to its "disease-specific generator". Although the "disease-specific generator" achieves a higher DS, it can only generate reports with diseases, leading to zero TN and thus a zero score on DOR. Such a model is not clinically useful as it lacks the ability to distinguish between normal and disease images. Introducing a classifier can alleviate this problem, but it gives rise to another issue of having FN, which is beyond the scope of this paper. Lastly, we show the discrepancy between the common metric and the proposed one in the last row by recording a second R2Gen model that is selected based on the best BLEU-4 score. Although this BLEU-based model gains an improvement of 0.1656 (the leading performance under this metric on IU-Xray) in our experiments, which are not presented in this table, it achieves close to zero in both DS and sensitivity. This implies that the majority of the generated reports do not align with the actual diseases, reducing their usefulness in a clinical setting.

#### 5.2.2. QUALITATIVE RESULTS.

Figure 3 provides a qualitative analysis that demonstrates the clinical efficacy of our methods and metrics. The generated

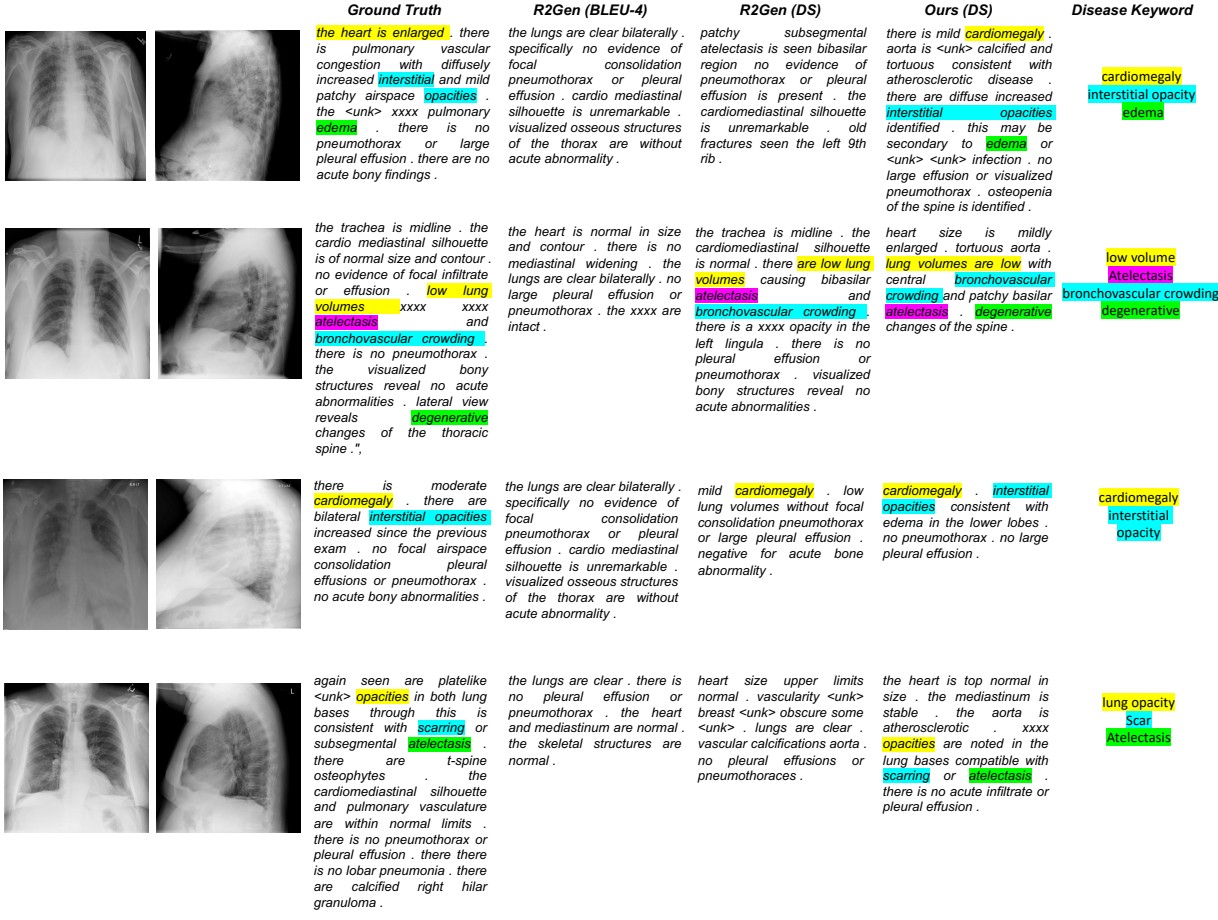

*Figure 3.* Qualitative comparison on abnormal cases. Only our method based on DS evaluation metric successfully generates the correct disease mentions. Highlighted words can be accurately captured by "Disease Keywords" from our KG.

reports reveal an important finding: the best R2Gen model, when selected based on the BLEU-4 metric (referring to as R2Gen (BLEU-4)), fails to generate disease-specific sentences, disregarding clinical-relevant information. In contrast, when selecting the models using our proposed DS metric (referring to as R2Gen (DS)), the chosen R2Gen model performs much better, indicating its ability to generate disease-specific sentences and emphasizing the need for a more clinical-relevant evaluation metric. Moreover, our two-stage generation approach, incorporating our augmentation strategy based on the DS metric, denoted as "Ours (DS)", effectively tackles the long-tailed issue by successfully capturing rare abnormalities such as "interstitial opacity" and "edema". The accurate descriptions of diseases generated by our approach, which align with the keywords in our knowledge graph (referred to as "Disease Keyword"), further validate the utility of our approach.

## 6. Conclusion

In this paper, we present the construction of a comprehensive knowledge graph focusing on chest X-ray images to uncover disease relationships and investigates the significance of disease mentions in medical report generation task. We propose a two-stage generation approach and a KG-based augmentation strategy to mitigate the challenges associated with imbalanced data sets. The KG developed in this study can be extended and utilized by other researchers. Furthermore, a novel evaluation metric is devised, leveraging the information captured in the KG to measure clinical relevance. This work serves as a catalyst for future exploration of the clinical efficacy in medical report generation.

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
