# OpenReview forum: "Rethinking Medical Report Generation: Disease Revealing Enhancement with Knowledge Graph"
_ICML.cc/2023/Workshop/IMLH — IMLH 2023 Poster_

### Official Review · Reviewer_UGwn · 2023-06-05
**Rethinking Medical Report Generation: Disease Revealing Enhancement with Knowledge Graph**

**Rating:** 8
**Confidence:** 4

**Review:**

In this paper, the authors propose a two-stage medical report generation approach. First, an image classifier is used to distinguish whether the input image shows diseases or abnormalities. Then the corresponding generator is chosen to generate the diagnostic report in the second stage. Furthermore, a data augmentation method based on a complete knowledge graph to mitigate the disease imbalance issue. In the experiments, the authors introduce a new evaluation metric, Diverse Sensitivity, to evaluate the performance of the proposed method. The experimental results on the UI X-ray dataset show that the proposed method achieves better performance than the baseline.
1.	Although a new evaluation metric is proposed, traditional evaluation metrics (such as BLEU) should be provided for reference.
2.	The authors extend the knowledge graph by adding more diseases and disease categories based on IU-Xray and MIMIC-CXR. But the improvement of the method using the new knowledge graph is not clear. The results of the model with the original knowledge graph should be provided for comparison.
3.	Only R2Gen is used as the baseline for comparison. It is desired to compare the proposed approach with more state-of-the-art approaches using the knowledge graph.

---

### Official Review · Reviewer_ezk9 · 2023-06-11
**Review from Reviewer ezk9**

**Rating:** 5
**Confidence:** 4

**Review:**

The focus of this paper, medical report generation, is very relevant to our venue. I like the idea of utilizing KGs to help with medical report generation. However, the paper lacks important explanations for audiences to understand their technical contribution and experimental results.

**Strengths**
- The authors investigate efforts to construct a disease-centric KG to help medical report generation.

**Weaknesses**
- There is no task definition in the paper. What exactly is medical report generation? Does the report purely generate from the input image? Sec 3.1 somehow mentions the process of medical report generation of R2Gen. A clear definition is recommended to make the paper more accessible.
- I am having trouble understanding the dataset, IU X-ray, in Sec 5.1. The authors mention that IU Xray contains 7470 images and 3955 reports. What are the relations between images and reports, given they have different numbers?
- It seems counter-intuitive to me that the data augmentation method can mitigate long-tail disease problems. From Fig.1, I observe that the frequent disease is even more dominant, after data augmentation. Can authors share more details about it?

---

### Meta-Review · Area_Chair_L67J · 2023-06-19

**Recommendation:** Accept (Poster)
**Confidence:** 5

**Metareview:**

Reviewers are generally positive in recommending the acceptance of this manuscript but also raise concerns. Please address them in the final version.

---

### Decision · Program_Chairs · 2023-06-20

Accept (Poster)